# Differences in Time Perspectives Measured under the Dramatically Changing Socioeconomic Conditions during the Ukrainian Political Crises in 2014/2015

**DOI:** 10.3390/ijerph19127465

**Published:** 2022-06-17

**Authors:** Oksana Senyk, Volodymyr Abramov, Viktoriia Bedan, Alina Bunas, Marharyta Hrechkosii, Olena Lutsenko, Tetiana Mandzyk, Marc Wittmann

**Affiliations:** 1Department of Psychology and Psychotherapy, Ukrainian Catholic University, 79000 Lviv, Ukraine; 2Faculty of Psychology, Taras Shevchenko National University of Kyiv, 01033 Kyiv, Ukraine; abramov.vladimir@gmail.com; 3Faculty of Psychology, National University “Odesa Law Academy”, 65000 Odesa, Ukraine; beviko@gmail.com; 4Department of Psychology, Oles Honchar Dnipro National University, 49000 Dnipro, Ukraine; bunasalina@gmail.com; 5Department of General and Developmental Psychology, Odessa I. I. Mechnikov National University, 65000 Odessa, Ukraine; rugytska@gmail.com; 6Department of Applied Psychology, V. N. Karazin Kharkiv National University, 61000 Kharkiv, Ukraine; olena.lutsenko@karazin.ua; 7Department of Psychology, Ivan Franko National University of Lviv, 79000 Lviv, Ukraine; t.m.mandzyk@gmail.com; 8Institute for Frontier Areas in Psychology and Mental Health, 79098 Freiburg, Germany; wittmann@igpp.de

**Keywords:** time perspective, time orientation, socioeconomic crisis, military crisis, political views

## Abstract

The characteristics of the individual’s time perspective in relation to changes in social, economic, and political conditions are of major conceptual interest. We assessed the time orientations of 1588 Ukrainian students living in two different regions (western and south-eastern Ukraine) with the Zimbardo Time Perspective Inventory (ZTPI) before (2010–2013) and during (2014–2016) the socioeconomic, political, and military crises which started in 2014, eight years before the war in 2022. We applied ANOVAs with the ZTPI dimensions as dependent variables and the period of testing (precrisis, postcrisis) as an independent variable for the two Ukrainian regions separately. The time perspectives of residents in the region most distant from the war zone (western), who positively assessed the change in the political situation around 2014, increased in the *future* time orientation and decreased in the *present-fatalistic*, *past-positive*, and 333 *present-hedonistic* time orientations. The time perspectives of residents in the regions closest to the war zone (southeastern) decreased in the *future* and increased in the *past-negative* and *present-fatalistic* time orientations, reflecting their negative judgments of the events. It is not the crisis itself, but the specific social, economic, and political factors and evaluations which define the time perspectives, which are flexible and adjust to changes during extreme life circumstances.

## 1. Introduction

Since the publication of the Zimbardo Time Perspective Inventory (ZTPI) in 1999, hundreds of studies on time-perspective (TP) research have been conducted using this tool. The TP construct is conceptualized as a “…nonconscious process whereby the continual flows of personal and social experiences are assigned to temporal categories, or time frames, that help to give order, coherence, and meaning to those events.” [1] (p. 1271), [2]. Time perspective is understood as the unique pattern which helps us assess and categorize our life experiences and is strongly related to every possible life domain, including psychological well-being, personal and professional achievements, and personality traits [3,4,5,6,7,8]. Taking this into consideration, it is important to find out whether it is a stable personality trait or can be changed, i.e., to improve one’s life situation [9]. We are familiar with the recent literature criticizing TP theory and its shortcomings [10,11]. The ZTPI was the only inventory measuring TP in the Ukrainian language available at the time of the study. Our study complements the enormous amount of empirical research applying the ZTPI in studies in different languages and cultures [12].

According to cognitive social learning theories, we first learn to reflect on our past events, plan for the future, or assess the current situation from our parents or other significant adults [13,14,15,16,17]. The important role of the family and its socioeconomic status on one’s time-perspective development is well documented in research on children who grow up in the deprived environment of an orphanage or in a socioeconomically deprived family. Such conditions often lead to a time perspective which is more biased towards the present, while the future time orientation is underdeveloped [18,19,20].

The family’s socioeconomic status has been shown to influence the development of a future time orientation in adolescents. The subjective orientation towards the future depends on the feasibility of plans made and the level of endorsement from a certain culture with its particular characteristics [21,22,23,24,25,26]. For example, Seginer and Lens [27] discuss how the endorsement of cultural demands defines the strength of the future time orientation in the domain of education for adolescent girls in Israel. A comprehensive review by Fieulaine and Apostoloidis [28] showed that a privileged socioeconomic status in adulthood is linked to more pronounced past and future time orientations and more positive attitudes towards them compared to individuals with a lower socioeconomic status. According to the authors, focusing on the present orientation in individuals with a lower socioeconomic status may be an adaptive strategy to cope with disadvantageous situations during crises and insecurity when the future is uncertain [28,29].

These studies indicate that the individual’s time perspective is strongly rooted in the social context of personal life experiences. There is clear supportive evidence that the characteristics of a person’s background culture and socioeconomic status shape an individual’s time perspective according to situational demands and the possibility of future rewards. An individual’s time perspective should, in principle, be modifiable in line with a profile which is optimal for psychological well-being and effective functioning. However, the simple fact that the time perspective is connected to different social factors does not mean that the impact of these factors can be easily outweighed. Attitudes towards time and behavioral patterns learned in early childhood and practiced according to personal experience for decades are difficult to change. This suggests that the time perspective may be a relatively stable individual characteristic throughout one’s lifetime.

How stable is an individual’s time perspective? Many studies supporting the time perspective as a stable individual characteristic have shown strong correlations between specific time orientations and different personality traits [30,31,32]. The different time-perspective dimensions (as measured by the ZTPI) are linked to all of the ‘big five’ personality traits (openness to experience, conscientiousness, extraversion, agreeableness, and neuroticism), as well as to locus of control, optimism, self-efficacy, aggression, impulsivity (especially sensation seeking), and many more [32,33,34,35,36], indicating the construct validity of the time perspective. One of the first attempts to assess the stability of time perspective was made by Luyckx and colleagues [37], who studied the time-perspective dynamics of freshman students over a four-month interval. The time perspective and an individual’s self-identity formation mutually reinforced one another, which led to significant changes in both constructs among those young people after a short period of time. Quite different results were found by Earl and colleagues [38]. They tested 367 retired Australians, all of whom completed the ZTPI three times, with two nine-month intervals in between. There was no strong modulation in the five ZTPI scales over the course of the 18 months, pointing to the stability of the time perspective construct in elderly individuals. The data were collected under similar national and global economic circumstances all three times [38], indicating that there were no significant external forces that might have led to changes in the time-perspective profiles.

A different attempt was undertaken by Wiberg and colleagues [39], who investigated the stability of the balanced time perspective (BTP), the ideal combination of time orientations that enable one to flexibly switch among different time dimensions according to personal needs and situational demands [6,8,36,40]. Seven participants were tested with the BTP profile again after a year and a half. Four of them had a stable BTP, whereas the time perspective of the other three had changed. One participant had increased his level of balance, and the profiles of two others indicated a decrease in their levels of balance [39]. Although the small number of participants makes it difficult to generalize the findings, this study pointed out important questions about how to measure the stability of the time perspective, namely by comparing the separate time orientations over a particular time interval and by exploring the dynamics of the whole system of the time perspectives.

Important for our study, it is known that the TP is changed in people suffering from loss and migration. In Syrian refugees in Greece, an increased past-negative and present-fatalistic and a decreased future time perspective was associated with post-traumatic stress disorder [41]. These three time orientations correlate similarly with people’s general life satisfaction [42]. The present study explored whether time perspectives change significantly under radical modifications in life circumstances, like political and economic factors. As shown in earlier studies, the time perspective correlates with a person’s social and economic status [29,43]. We also assumed that significant changes in the social and economic living conditions would provoke a change in time perspectives.

The time perspective of Ukrainian students was observed before and during the period of profound national, social, economic, and political crises starting in 2014. The pre-crisis period was characterized by relative social, economic, and political stability. The crisis period was marked by a high level of social, economic, and political turbulence: the national currency rapidly depreciated by a factor of three, an administrative region was annexed, two other administrative regions were isolated by the war line, and the level of unemployment increased greatly. All these factors caused huge waves of mass migration from the annexed regions to other Ukrainian regions and abroad. According to official numbers, more than a million Ukrainians left their homes during the first two years of the crisis of 2014. Differences in social, economic, and political characteristics between the pre-crisis and crisis periods were sufficiently significant to expect changes in the residents’ time perspectives if the TP construct is actually sensitive to situational factors. The TP data analyzed in the study were gathered by Ukrainian researchers from early 2010 to 2018 as part of an attempt to collect norm data of the ZTPI for Ukraine (see below). After the beginning of the crisis, a cross-sectional design of the study with two time points was applied to investigate whether there were any differences in time perspectives measured under different socioeconomic circumstances in two different Ukrainian regions. We were unable to implement a strict longitudinal design with intra-individual measurements across two time points. What happened in Ukraine could not be anticipated, which is why the presented study could not be planned in advance. The TP data analyzed in the study were gathered by Ukrainian researchers from early 2010 to 2018.

## 2. Materials and Methods

### 2.1. Participants

A sample of 1588 individual students participated in a series of studies in different universities in Ukraine. Two regional sub-samples were formed: (1) 1037 residents from the Lviv region, the most western Ukrainian region and the one most distant from the war zone (Table 1); (2) 551 residents of the most endangered regions, the eastern Ukrainian regions closest to the war zone and the southern coastal region, which has been plagued by a highly unstable internal situation since the beginning of the crisis and was therefore psychologically similar to those close to the armed conflict (Table 2).

The decision to analyze the samples separately was based not only on the proximity to the war zone and the internal socio-political situations in the regions, but also on their residents’ political views. According to sociological research conducted during the pre-crisis period and after the crisis had begun, up to 90% of western residents supported the changes initiated at the beginning of the crisis, while south-eastern Ukrainian regions had quite opposite views. The majority of these residents (about 70%) were clearly against the ongoing changes and named the revolution one of the most negative events in Ukrainian history (Democratic Initiatives Foundation) [44,45,46]. Since the time perspective reflects individual views of one’s own past, present, and future [1,47] and is connected to one’s life situation, the different perceptions and views of the ongoing sociopolitical situation created an important distinguishing factor to analyze the data from different regions separately.

The sample for the western Ukrainian region was comprised of students from two national universities (Ivan Franko National University of Lviv and the Lviv Polytechnic National University) who were residents of the Lviv region. The total sample consisted of 1037 students (41% males and 49% females; the gender of 10% of the participants was unknown because that information was taken from studies which did not include such information). A total of 432 students were questioned in 2010–2011 and 116 in 2012–2013 (pre-crisis), and 489 students comprised the “crisis” group that was collected in 2015–2016. We decided to distinguish two pre-crisis subgroups, since the interval between data collections (1 year 8 months) was almost equal to that between the second pre-crisis subgroup and the crisis group (1 year 4 months). The surveys contributing data to this study sample were performed in group settings. More detailed characteristics of the sample are presented in Table 1.

The sample for the south-eastern Ukrainian region consisted of students from different universities located in the Dnipro, Kharkiv, and Odessa regions. A total of 154 were male and 397 female, with a mean age of 19.05 years. A total of 279 were questioned in 2013 (the pre-crisis year) and 272 during the years of 2014–2015 (after the start of the armed conflict, which marked the beginning of socioeconomic and political crisis). More detailed characteristics of the sample are presented in Table 2.

Considering the above-mentioned assumptions about the regional characteristics, we decided to analyze the time perspectives separately by regions, assuming that different views of the sociopolitical situations and different proximities to the war zone would result in different time perspectives between the pre-crisis and crisis periods.

The decision to explore the time-perspectives separately by regions was based on three premises: (1) the objective severity of the exposure to danger due to the proximity to the war zone or internal instability of the region; (2) the regional differences in the residents’ political views, which represented a portion of their overall subjective views underlying individual time perspectives; and (3) the significant differences in time orientations between regions during the pre-crisis period. The independent variable of interest was the period of testing.

### 2.2. Instrument

The participants completed a form on their age, sex, and place of residence, as well as the Zimbardo Time Perspective Inventory in its Ukrainian or Russian adaptation, depending on their native language (Ukrainian adaptation by Senyk [48]; Russian adaptation by [49]). The results were calculated according to updated keys for Ukrainian and Russian versions of the ZTPI (validated on Ukrainian- and Russian-speaking Ukrainians) [50]. Participants responded to each of the 56 items on a 5-point Likert scale (1 = very untrue of me; 5 = very true of me). The results were calculated in accordance with the updated keys for the Ukrainian version of the ZTPI [50], which slightly differs from the first version in the present-fatalistic scale.

The inventory itself measures five dimensions of the time perspective. The past-negative scale reflects a generally negative, aversive view of one’s own past. Due to the reconstructive character of the past, such negative attitudes could reflect real experiences of negative or traumatic moments in the past, a negative reconstruction of an actually not-so-aversive past, or a combination of both. The present-hedonistic scale reflects a hedonistic, risk-taking attitude toward life and presupposes enjoying the present moment with little concern for the further consequences of one’s behavior. The future scale measures a general future orientation, which suggests that behavior is dominated by the effort made to achieve the goals set and possible rewards in the future. The past-positive scale relates to fond and sentimental attitudes towards the past, when past experience and times are remembered as something pleasant, with a tendency towards nostalgia. The present-fatalistic scale reveals a fatalistic, helpless, and hopeless attitude towards the future and life in general; individuals with such a time orientation believe in fate and are certain that they cannot influence present or future events in their lives [1].

## 3. Results

First, we compared time orientations from the pre-crisis period in the western region and in the south-eastern regions. The Student’s t-test showed significant differences in present-hedonistic and future time orientations (t = −7.34 and t = −3.32 respectively, *p* < 0.001) between regions. The hedonistic and future time orientations were more pronounced in the eastern regions (compare values in Table A1 and Table A2 in the Appendix B). In the post-crisis period, south-eastern regions scored significantly higher on both negative and positive past orientations (t = −4.63 and t = −4.00, respectively, *p* < 0.001), and significantly higher on both hedonistic and fatalistic present orientations (t = −9.20 and t = −7.16, respectively, *p* < 0.001), while showing no difference in future time orientation, as compared to the western region (Table A1 and Table A2 in the Appendix B).

Then, we applied ANOVAs to analyze the variance in each time orientation separately across time and for region and controlled for gender.

### 3.1. Western Region

Figure 1 shows that there were no significant differences in time-orientation scores between the first and the second pre-crisis periods; the main change in time perspective was observed in the third period, which was characterized by the socioeconomic crisis. The future time orientation increased during the crisis period, while the scores on the present-hedonistic and past-positive scales decreased. There was a decrease in the scores on the present-fatalistic scale throughout all three periods. No dynamics were observed for the past-negative time orientation.

The two pre-crisis subgroups were united into one group of 548 participants and then again compared with the crisis group (N = 489). Following separate ANOVAs, the future time orientation was significantly higher in the crises period as compared to the pre-crisis period (F(1, 929) = 10.88, *p* = 0.001, ηp^2^ = 0.012). The present fatalistic (F(1, 929) = 11.87, *p* = 0.001, ηp^2^ = 0.013), the present-hedonistic (F(1, 929) = 28.57, *p* < 0.001, ηp^2^ = 0.030), and the past-positive (F(1, 929) = 27.46, *p* < 0.001, ηp^2^ = 0.029) orientations were lower during the crisis as compared to the pre-crisis period. No difference for the past-negative time orientation was found over time (F(1, 929) = 0.18, *p* = 0.668, ηp^2^ < 0.001). The mean values for time orientations in the pre-crisis and crisis groups can be found in Appendix B.

### 3.2. South-Eastern Region

Figure 2 shows that both past-negative and present-fatalistic time orientations were higher during the crisis period, while the scores of future time orientation were lower compared to the pre-crisis period. The mean scores for time orientations in the pre-crisis and crisis groups can be seen in the Appendix B. The variance for each time orientation was separately calculated with ANOVAs. This revealed significantly higher values in the past-negative (F(1, 547) = 16.17, *p* < 0.001, ηp^2^ = 0.029) and present-fatalistic time orientation (F(1, 547) = 25.63, *p* < 0.001, ηp^2^ = 0.045) after the onset of the crisis as compared to before. Values for the future time orientation (F(1, 547) = 6.70, *p* = 0.010, ηp^2^ = 0.012) were significantly lower during the crises than before the crises. No differences for present-hedonistic and past-positive time orientations were identified (F(1, 547) = 1.09, *p* = 0.298, ηp^2^ = 0.002, and F(1, 547) = 3.09, *p* = 0.079, ηp^2^ = 0.006, respectively).

## 4. Discussion

The present study examined variations in Ukrainian youths’ time perspectives measured under the different socioeconomic and political conditions prevailing during the pre- and post-crises surrounding the year 2014, eight years before the war started in 2022. Stated in longitudinal terms, the time perspectives of the Ukrainian youth shifted towards a decrease in the future and an increase in the past-negative and present-fatalistic time orientations with the beginning of the socioeconomic crisis in the most unstable and closest regions to the war zone (south-eastern). These findings coincide with the general concept of time perspective, which states that the future time orientation decreases in times of material or psychological deprivation, whereas the present time orientation becomes more pronounced [29,43,51]. Such a change in time perspective helps to effectively adapt to the new circumstances when the distant outcomes are impossible to anticipate and novel challenges need to be dealt with [29]. In the region most distant from the armed conflict in 2014/2015, the western Ukrainian region of Lviv, the direction of the change in time perspective was the opposite. After the crisis had started, an increase in the future, a decrease in the present-hedonistic and present-fatalistic time orientations was observed. The past-positive orientation also decreased.

Apart from the size of the identified effect, another argument for the validity of the findings is their consistency. No difference was found between the two pre-crisis periods, although the time interval between them was almost equal to the time interval between the second pre-crisis and the crisis periods (1 year 8 months and 1 year 4 months, respectively). The two different pre-crisis periods in the western-region sample did not show a significant difference in time orientations. We can conclude that not the time interval per se contributes to the differences in time perspective, but the visible changes in life circumstances between different time intervals. One of the functions of the time perspective is to categorize one’s personal and social experience [1]. If the social circumstances change enough to notably influence personal experience, the time perspective adapts accordingly, whereas it does not change significantly during stable conditions.

This conclusion coincides with the findings of Luyckx et al. [37] and Earl et al. [38]. In their longitudinal study, Luyckx et al. [37] showed that the time perspective changed significantly under the intense influences of social experience, even after a comparably short interval. Freshman college students showed changes in time perspectives as they shifted towards an increase in the future and a decrease in the present time orientations after just four months. The authors explained that the dynamics of time perspective were due to a new social role of college students, who were in the process of preparing themselves for their careers and future adult lives [37].

The study by Earl et al. [38] showed that the time perspective does not change significantly if measured under stable conditions, even after a year and a half. The authors conducted three measurements of the time perspectives of 367 retired individuals at nine-month intervals, each under similar global economic circumstances. Finding no changes over time, Earl et al. [38] concluded that time perspectives are difficult to change. We, however, argue that significant changes in time perspectives do occur only if visible changes in personal or social experience happen.

The question is why different and even opposite directions in time-perspective changes were observed in the two Ukrainian regions. The future orientation decreased, and past-negative and present-fatalistic time orientations increased in the south-eastern regions. In the western region, the future time orientation increased, and the present-fatalistic, present-hedonistic, and past-positive time orientations decreased. One possible answer lies in the economic situation. People closer to the war zone in the south-eastern regions might have experienced more severe consequences of the economic decline, whereas almost all economic sectors in the western region remained unchanged, including tourism from abroad.

The proximity of south-eastern regions to the war zone might also have undermined the basic safety of the inhabitants. This could account for the south-eastern residents’ shift in time perspectives towards an increase in the past-negative and present-fatalistic and a decrease in the future time orientations. When life becomes endangered by forces one cannot control, the time perspective adapts by increasing in fatalistic attitudes towards the present and by decreasing the future outlook, which no longer makes any sense due to its total unpredictability.

A further influence on the time perspective may have been expectations of the political developments. As mentioned in the introduction, the views of the crisis and its political ramifications differed between residents in the western and the south-eastern Ukrainian regions. The western regions were characterized by mostly positive attitudes towards the events that preceded the crisis and their political consequences, expressing belief in a better future in accordance with the sociological survey (Democratic Initiatives Foundation) [46]. This corresponds to the identified increase in the future time orientation in the residents of the western region. The majority of residents of the south-eastern regions had mostly negative views of those events (Democratic Initiatives Foundation) [46], which corresponds to the revealed increase in present-fatalistic and past-negative time orientations.

Our study shows that time perspectives change significantly due to notable changes in social, economic, and political processes. However, it is probably not the change in the situation itself, but the specific combination of factors and perceptions of them, that influences the time perspective. Among various predictors, economics, safety needs, and political preferences played a role in the observed differences in time perspectives measured before and after the start of the socioeconomic, political, and military crisis.

The strength of this study is the large data set, which allowed us to assess differences in time perspectives measured under diverse socioeconomic and political conditions over time. However, there are also limitations. The data used in this study stemmed from different surveys with the aim to collect norm data of the ZTPI for Ukraine. It was impossible to plan the study in advance; as a result, a strictly longitudinal within-subject design was not possible. Therefore, any causal interpretations concerning the period of testing on the time perspective should be treated with caution. The changes in time perspectives due to changes in social, economic, and political conditions are suggestive, but longitudinal, within-subject studies are still needed to complement our findings. Another limitation is the fact that we did not include additional information about participants’ subjective views of the ongoing situation. We could not examine whether there really were any clearly significant correlations between time perspectives and political attitudes. All we could rely on were the results of corresponding sociological surveys. We referred to the average population data by assigning students’ time perspectives to average sociological, economic, and political indices in a region. A future study should include questions about feelings and judgments of personal welfare, safety, and personal expectations, as well as expectations of the country’s political future. Finally, the study sample was comprised of students only and was characterized by a predominance of women, which could have impacted the results. Advanced planning and including different age groups and people from different social strata would increase the credibility of the research. Social, economic, and political turmoil cannot be easily anticipated, especially not for the sake of psychological studies. Even now, in 2022, the war took most people by surprise. We relied on serendipitous data, which we analyzed to the best of our knowledge.

## 5. Conclusions

The article examined differences in Ukrainian youths’ time perspectives measured under diverse socioeconomic conditions. The time perspectives tested during the social, economic, and political crises in 2014/2015 significantly varied from the ones previously measured under stable circumstances. The revealed differences did not fall into one pattern, but they depended on the specific characteristics of the crisis and individual perceptions of it. In the southeastern regions, which suffered from severe consequences of the crisis, the time perspectives shifted towards a decrease in future and an increase in past-negative and present-fatalistic time orientations. In the western regions, where the crisis situation was characterized by a predominantly positive political attitude and was not as severely affected socioeconomically, the observed shift in the time perspectives took a positive direction, with a decreased emphasis on the negative present time orientation and an increase in the future orientation. The results of the presented study support the time perspective as a dynamic system which adjusts to visible changes in socioeconomic and political characteristics of the living situation.

## Figures and Tables

**Figure 1 ijerph-19-07465-f001:**
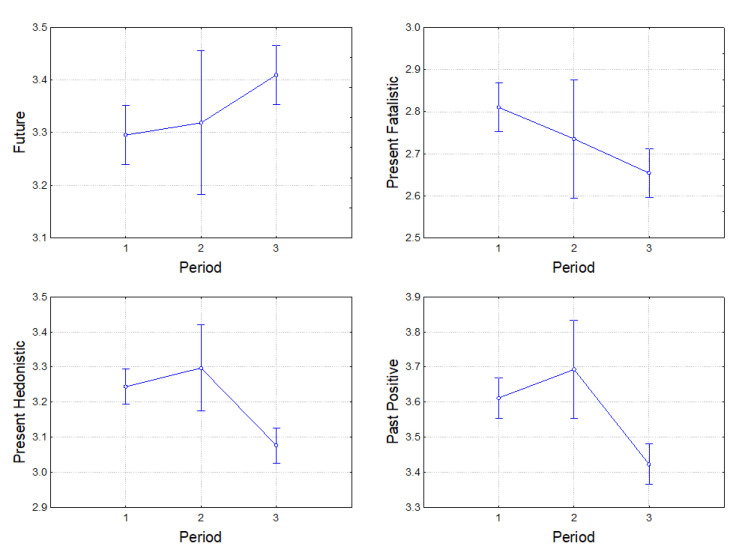
Graphs of significant differences in the *future*, *present fatalistic*, *present-hedonistic*, and *past-positive* time orientations observed from pre-crisis to crisis periods in the Lviv region. Axis “period”: 1—pre-crisis subgroup 1 (June 2010–March 2011), 2—pre-crisis subgroup 2 (December 2012–October 2013), 3—crisis group (March 2015–April 2016).

**Figure 2 ijerph-19-07465-f002:**
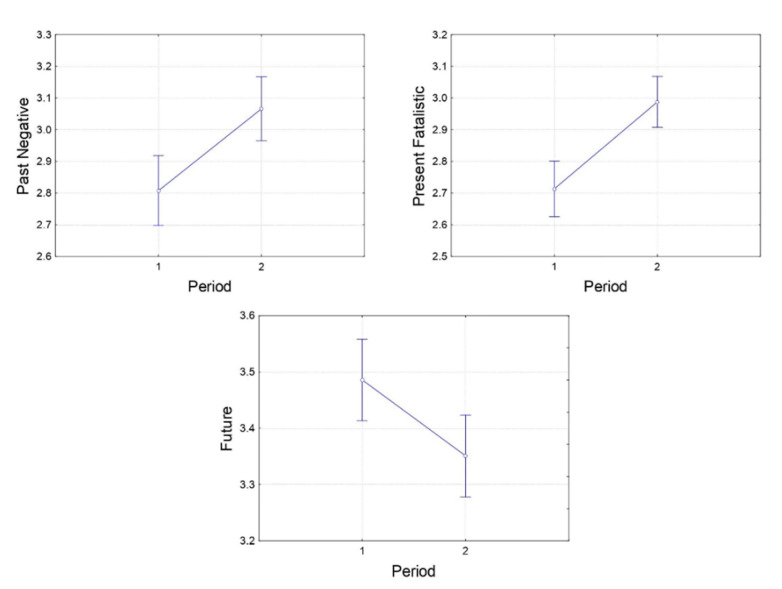
Graphs of significant differences in the *past-negative*, *present fatalistic*, and *future* time orientations observed from pre-crisis to crisis periods in the south-eastern Ukrainian regions. Axis “period”: 1—pre-crisis subgroup 1 (March 2013–November 2013), 2—crisis group (April 2014–May 2015).

**Table 1 ijerph-19-07465-t001:** Characteristics of the western (Lviv) region sample based on age, sex, and time of data collection.

	Period	Number	Mean Age [Years]	Males N	Females N	Gender Unknown N
1	**Pre-crisis subgroup 1** June 2010–March 2011	432	19.58 SD = 1.72	189 44%	243 56%	0
2	**Pre-crisis subgroup 2** December 2012–October 2013	116	19.18 SD = 1.52	33 28%	39 34%	44 38%
3	**Crisis group** March 2015–April 2016	489	20.19 SD = 1.93	200 41%	229 47%	60 12%
4	**Total** June 2010–April 2016	1037	19.83 SD = 1.84	422 41%	511 49%	104 10%

**Table 2 ijerph-19-07465-t002:** Characteristics of the south-eastern region sample based on age, sex, and time of data collection.

	Period	Number	Mean Age [Years]	Males N	Females N	Gender Unknown N
1	**Pre-crisis** March 2013–November 2013	279	19.05 SD = 1.34	77 28%	202 72%	0
2	**Crisis** April 2014–May 2015	272	19.19 SD = 1.71	77 28%	195 72%	0
3	**Total** March 2013–May 2015	551	19.35 SD = 1.76	154 28%	397 72%	0

## Data Availability

The data presented in this study are available in the Appendix A here.

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
