# Peer review of "Differences in Time Perspectives Measured under the Dramatically Changing Socioeconomic Conditions during the Ukrainian Political Crises in 2014/2015"

_ijerph, 2022, doi:10.3390/ijerph19127465_

Round 1

Reviewer 1 Report

This is an interesting and quite well-written paper concerning time perspectives before and during the crisis in Ukraine. Although the paper has some merits, I cannot recommend its publication in its current form.

My major concern is a methodological one, and it is related to not including possible predictors (or consequences) of time perspectives. If possible, it would be recommended that the Authors include other variables in the analyses and test their predictive role in time perspectives before and during the crisis. Without such analysis, the value of the paper is very limited.

Minor comments:

- “After balancing the sample by age and gender….” I am curious about what method the Authors used to balance the sample. This should be clearly stated in the manuscript.

- The Authors mentioned that the model was controlled for gender. If any control variables were included in the model, it is the ANCOVA model (analysis of covariance) rather than ANOVA.

- Taking into account the relationships between age and time perspectives (Laureiro-Martinez, Trujillo, & Unda, 2017), the model should also be controlled for age. Moreover, if the Authors measure religiosity, they are encouraged to consider adding this variable as a control variable since there is substantial evidence of its relationship with time perspectives (Łowicki, Witowska, Zajenkowski, & Stolarski, 2018). In addition, the sample consisted solely of students. Presumably, some of them were not only studying but also working, which may influence their financial situation. Therefore, I recommend controlling working status and/or financial situation (if these variables were measured). Moreover, did the Authors measure the level of perceived socio-economic insecurity? This variable may explain some degree of variance in time perspectives before and during the crisis.

- The results sections should be rewritten. Since the study was not longitudinal, using such verbs as “increase” or “decrease” is unjustified and should be replaced by “higher levels” or “lower levels.”

References

Laureiro-Martinez, D., Trujillo, C. A., & Unda, J. (2017). Time perspective and age: A review of age associated differences. Frontiers in Psychology, 8: 101. https://doi.org/10.3389/fpsyg.2017.00101

Łowicki, P., Witowska, J., Zajenkowski, M., & Stolarski, M. (2018). Time to believe: Disentangling the complex associations between time perspective and religiosity. Personality and Individual Differences, 134, 97–106. https://doi.org/10.1016/j.paid.2018.06.001

Author Response

Rev 1: This is an interesting and quite well-written paper concerning time perspectives before and during the crisis in Ukraine. Although the paper has some merits, I cannot recommend its publication in its current form.

My major concern is a methodological one, and it is related to not including possible predictors (or consequences) of time perspectives. If possible, it would be recommended that the Authors include other variables in the analyses and test their predictive role in time perspectives before and during the crisis. Without such analysis, the value of the paper is very limited.

Answer: We want to emphasize that the Ukrainian authors were collecting norm data for the newly developed Ukrainian version of the ZTPI over the years. Therefore, the large number of 1588 Ukrainian students enclosed in the study. Since this norm study had the sole purpose of collecting data with the ZTPI we did not include other questionnaires as would be done in typical studies. Of course, we cannot infer the predictive roles of various potential variables in time perspective in a quantitative way, but we think that the data presented is very valuable as it shows the changes of a personality trait due to a traumatic political event.

Rev. 1: Minor comments:

- “After balancing the sample by age and gender….” I am curious about what method the Authors used to balance the sample. This should be clearly stated in the manuscript.

Answer: We had an age range of participants as between 17 and 24 with a mean of 19.62 (SD = 1.82) across all subjects (the student population). As can be seen in table 1 and 2, participants in the subgroups were all on average between 19 to 20 years of age. We deleted the misleading sentence and replaced it by a more meaningful one.

Regarding our convenience sample, there were some differences in the percentage of males and females, also in the number of unknown gender responses. See below for the statistical control.

Rev. 1: The Authors mentioned that the model was controlled for gender. If any control variables were included in the model, it is the ANCOVA model (analysis of covariance) rather than ANOVA.

Answer: This is an important point. According to statistical literature, the goal of an ANOVA is to explain (to model) as much variation in a continuous variable as possible, by using one or more categorical variables to predict the variation (Elmore, 2020). An ANCOVA requires a continuous (scale/interval/ratio) dependent variable, and categorical factors and scale covariates as independent variables. So the common applications for ANCOVA are similar to ANOVA (to detect a difference in means of independent groups), while controlling for scale covariates which should be correlated with the dependent variable (Jamieson, 2004; Leppnik, 2018, Rothwell). Since in our study we did not have any continuous independent variables correlated with DV (time perspective) we did not apply an ANCOVA, but an ANOVA, including sex as a control variable, so that the variance caused by sex is subtracted from the main effect of interest which was the period (pre-crisis and crisis). That is, since an ANCOVA requires at least one continuous IV, which correlates with DV, we used the ANOVA.

Elmore, J.G., Wild, D.M.G., Nelson, H.D., Katz, D.L. (2020). Analyzing relationships between multiple variables. In Jekel's Epidemiology, Biostatistics, Preventive Medicine, and Public Health, pp. 172-181.

Jamieson, J. (2004). Analysis of covariance (ANCOVA) with difference scores. International Journal of Psychophysiology, 52(3), 277–283. https://doi.org/10.1016/j.ijpsycho.2003.12.009

Leppink, J. (2018). Analysis of Covariance (ANCOVA) vs. Moderated Regression (MODREG): Why the Interaction Matters. Health Professions Education, 4(3), 225–232. https://doi.org/10.1016/j.hpe.2018.04.001

Rothwell, J. ANCOVA (Analysis of Covariance). In Statstutor community Project by the University of Sheffield. Available at https://www.sheffield.ac.uk/polopoly_fs/1.531229!/file/MASH_ANCOVA_SPSS.pdf (accessed on 08.06.2022).

Rev. 1:  Taking into account the relationships between age and time perspectives (Laureiro-Martinez, Trujillo, & Unda, 2017), the model should also be controlled for age. Moreover, if the Authors measure religiosity, they are encouraged to consider adding this variable as a control variable since there is substantial evidence of its relationship with time perspectives (Łowicki, Witowska, Zajenkowski, & Stolarski, 2018). In addition, the sample consisted solely of students. Presumably, some of them were not only studying but also working, which may influence their financial situation. Therefore, I recommend controlling working status and/or financial situation (if these variables were measured). Moreover, did the Authors measure the level of perceived socio-economic insecurity? This variable may explain some degree of variance in time perspectives before and during the crisis.

Answer: These are fantastic ideas which would/could have been used in a typically planned study. That was not possible here where the crisis came to a surprise and could not be anticipated. We can only compensate with the dramatic changes we detect in different time orientations before vs. after the onset of the crisis, as assessed with the norm data. It was the political drama that changed the data from a typical norm study that would be used to classify individual responses to the ZTPI being a detector of psychological changes.

Rev 1: The results sections should be rewritten. Since the study was not longitudinal, using such verbs as “increase” or “decrease” is unjustified and should be replaced by “higher levels” or “lower levels.”

Answer: We have changed the wording in the results section to accord with the cross-sectional nature of the study design. In the discussion we keep the terms in longitudinal fashion as a means of generalization.

Reviewer 2 Report

The presented study is very important due to social context. The text is well stryctured and present in details all outcomes.

I would suggest to  be more precise to inform the readers about the date when the measurement was done both in the abstract ans in the Material and Method part.

The tome perspective evaluetion was done in two Ukrainian regions due to the socio-economic differences however these data are described without any comparison, absolutely separatelly. It would be more interesting to compare not only the time-points of the measurement but  also the regional data. I would suggest to expand your analysis to include this.

Author Response

Rev. 2: The presented study is very important due to social context. The text is well stryctured and present in details all outcomes.

I would suggest to  be more precise to inform the readers about the date when the measurement was done both in the abstract ans in the Material and Method part.

Answer: We provide the information of the exact measurement years of the study, as it is indicated in Tables 1 and 2, especially in the abstract.

Rev. 2: The tome perspective evaluetion was done in two Ukrainian regions due to the socio-economic differences however these data are described without any comparison, absolutely separatelly. It would be more interesting to compare not only the time-points of the measurement but  also the regional data. I would suggest to expand your analysis to include this.

Answer: In the results section (now moved to it), the following paragraph concerning the pre-crises period we already have the following paragraph:

We compared time orientations from the pre-crisis period in the western region and in the south-eastern regions. The student's t-test showed significant differences in present-hedonistic and future time orientations (t = -7.34 and t = -3.32 respectively, p < 0.001) between regions. The hedonistic and future time orientations were more pronounced in the eastern regions (compare values in Tables A1 and A2 in the Appendix 1).

We now add the statistics for the post-crisis period:

In the post-crisis period south-eastern regions scored significantly higher on both negative and positive past orientations (t = -4.63 and t = -4.00 respectively, p < 0.001), and significantly higher on both hedonistic and fatalistic present orientations (t = -9.20 and t = -7.16 respectively, p < 0.001), while showing no difference in future time orientation as compared to western region.

Round 2

Reviewer 1 Report

The Authors addressed all my concerns. I can now recommend the paper for publication.